# The Pathological Culprit of Neuropathic Skin Pain in Long COVID-19 Patients: A Case Series

**DOI:** 10.3390/jcm11154474

**Published:** 2022-07-31

**Authors:** Teresa Grieco, Vito Gomes, Alfredo Rossi, Carmen Cantisani, Maria Elisabetta Greco, Giovanni Rossi, Alvise Sernicola, Giovanni Pellacani

**Affiliations:** 1Department of Clinical Internal Anesthesiological and Cardiovascular Sciences, “Sapienza” University of Rome, Viale del Policlinico 155, 00161 Rome, Italy; teresa.grieco@uniroma1.it (T.G.); alfredo.rossi@uniroma1.it (A.R.); carmen.cantisani@uniroma1.it (C.C.); mariaelisabettagreco1@gmail.com (M.E.G.); giovanni.rossi@uniroma1.it (G.R.); giovanni.pellacani@uniroma1.it (G.P.); 2Department of Anatomy and Pathology, Ospedale San Filippo Neri, Via G. Martinotti 20, 00183 Rome, Italy; vito.gomes@aslroma1.it

**Keywords:** COVID-19, neuropathy, peripheral nervous system, SARS-CoV-2, paresthesia, inflammatory pain, chronic pain, dermatopathology, neurologic symptom

## Abstract

Cutaneous neurosensory symptoms have become increasingly reported findings in COVID-19; however, these virus-related manifestations are largely overlooked, and their pathology is poorly understood. Moreover, alterations of skin sensibility currently recognize no clear histopathology substrate. The purpose of this study was to provide pathology evidence of neurosensory skin system involvement in COVID-19 patients complaining of subjective neurological symptoms affecting the skin. Out of 142 patients, six long COVID-19 cases complaining of cutaneous subjective neurological symptoms assessed on an NTSS-6 questionnaire underwent histopathological and immunohistochemical analyses of skin areas affected by paroxysmal diffuse burning and itching sensations. Two patients also performed electroneurography examination. The histology investigation showed hypertrophic glomus vascular bodies with hypertrophic S100+ perineural sheath cells and adjacent hypertrophy of the nerve branches associated with increased basophil polysaccharide matrix. Electroneurography revealed disturbances of A-delta and C dermal neuronal fibers. The main limitation of this study consisted of a limited number of skin biopsy samples, requiring further investigation. Histopathology findings are consistent with hypertrophy of nerve endings, suggesting a condition such as “dermal hyperneury”, a recently reported small nerve hypertrophy condition affecting sensory C fibers. Such a neuropathic basis could explain dysesthesia experienced by the patients, as previously described in postherpetic neuralgia.

## 1. Introduction

Severe acute respiratory syndrome coronavirus 2 (SARS-CoV-2) typically manifests with interstitial pneumonia or with acute respiratory distress syndrome (ARDS) and multisystem involvement in severe cases. It is now established that SARS-CoV-2 is able to infect a wide variety of cells, including neurons, by exploiting the angiotensin-converting enzyme 2 (ACE2) entry receptor expressed almost ubiquitously in human tissues [1]. The coreceptor transmembrane protease serine 2 (TMPRSS2) has also been recognized [2]. Among the extrapulmonary manifestations of SARS-CoV-2, nervous system involvement constitutes an emerging topic in the recent literature. Central nervous system manifestations are currently well-characterized, while peripheral nervous system involvement still lacks in-depth understanding, despite growing published evidence. Ageusia and anosmia were the first subjective neurologic symptoms (sNS) reported in COVID-19 and are currently considered “hallmark” signs in the prodromal phase of infection [3]. Nervous interactions of SARS-CoV-2 involve direct axonal spreading along olfactory receptor neurons, through which SARS-CoV-2 gains access to the bloodstream, or retrograde neuronal transport, involving ACE-2 receptor mediated endocytosis of virions. Hypothetical neurotoxic mechanisms of SARS-CoV-2 could be related to autophagy, apoptosis, and necrosis [4]. Indeed, the World Federation of Neurology has previously described neurological manifestations related to COVID-19 in the central and peripheral nervous systems [5].

To this date, the etiology underlying COVID-19 neuropathies remains poorly understood, and histopathology is still lacking.

The purpose of this study was to investigate the cutaneous pathology underlying sNS of largely overlooked burning and itching skin sensations in long-COVID-19 patients. Histology and immunohistochemistry studies were performed on skin samples from each subject.

## 2. Materials and Methods

Out of 142 subjects who consulted our post-COVID-19 follow-up clinic (PCFC) of the Dermatology department in “Sapienza” University of Rome from May to September 2021, we enrolled six COVID-19 cases complaining of cutaneous sNS following SARS-CoV-2 infection (Table 1). All patients referred to peripheral dysesthesia not otherwise explained after other conditions such as drugs, diabetes, or neurological diseases that had been ruled out based on personal history. To characterize the neurologic disorder, an NTSS-6 questionnaire was administered. Two of our patients also underwent electroneurography (ENG) examination. Additionally, laboratory exams were routinely performed in all patients to rule out autoimmune conditions, with negative results.

Furthermore, skin biopsies of the burning involved areas were effectuated to perform histology and immunohistochemistry. 

Finally, all patients underwent follow up clinical visits after 3 and 6 six months.

## 3. Results

### 3.1. Case 1

A 49-year-old Caucasian woman consulted our PCFC complaining of cutaneous burning pain, arising bilaterally on the shoulders, extending on the lateral surface of both arms, and gradually becoming generalized. The skin examination showed red-purple erythema on the palmar region, bilaterally, coexistent with livedo reticularis on the legs. Livedoid skin manifestations appeared two weeks earlier and rapidly improved in the following days. Though the patient presented with bilateral erythema of the palms, the clinical suspect of erythromelalgia was excluded due to the absence of typical triggers for episodic pain and to the widespread burning sensation involving apparently normal distant skin. The positive serum antibody test against SARS-CoV-2 confirmed the diagnosis. The rheumatological and autoimmune vasculitis panel was negative as well. The histology examination of the right palmar skin showed glomus vascular bodies, hypertrophic S100+ peri-neural sheath cells, and adjacent hypertrophy of the nerve branches with increased basophil polysaccharide matrix (Figure 1). No inflammatory infiltrate was observed in the superficial or in the deep peri-adnexal areas. These pathologic findings strongly suggested a manifestation on a neuropathic basis.

### 3.2. Case 2

A 67-year-old Caucasian woman was referred to the PCFC following recovery. The patient complained of burning pain and an itching sensation, beginning in the late period of COVID-19 infection. Skin pain started on the chest, extending irregularly to the four limbs and reaching the palmar and plantar surfaces. The skin examination was negative. 

A biopsy was performed on the skin with burning pain in the abdominal area. Histology findings showed small nerve ramifications in the medium and deep dermis. Fibers appeared wavy and hypertrophic and displayed a slight increase in basophil mucopolysaccharide content. Intraepidermal S100+ dendritic cells were irregularly distributed (Figure 2). CD34+ interstitial cells/myofibroblasts and CD117+ melanophages were present in the superficial dermis (80× sqmm). 

### 3.3. Case 3

A 64-year-old Caucasian man was referred to our PCFC in December 2020 following the onset of a generalized cutaneous sensation of burning and itching skin. The patient contracted the SARS-CoV-2 infection 2 months before and was admitted to our hospital for pulmonary involvement with characteristic anosmia and dysgeusia. Acute symptoms resolved in 2 weeks, leaving asthenia, shortness of breath, and mental fog, indicating a long-COVID-19 condition. Dermatology was consulted for the management of the skin burning—paresthesia. The skin biopsy was performed on the skin in the abdominal region. Histology showed hypertrophic nerve bundles that were markedly visible in the mid dermis (Figure 3). Additional hypertrophic nerve bundles were evident in the lower dermis. No inflammatory reaction was present.

### 3.4. Case 4

A 35-year-old Caucasian woman presented to our PCFC complaining of the onset of hives in December 2020, following recovery from a COVID-19 pericarditis requiring hospitalization in September. The patient reported an intense persistent itching and stinging sensation. The skin biopsy was obtained from popliteal itchy skin. A glomeruloid cluster of nerve bundles with basophilic mucin surrounding the nerve sheath was present in the mid dermis without an inflammatory reaction (Figure 4). Other small nerve bundles were present in the mid and lower dermis. ENG reported reduced sensory nerve action potential amplitude; nerve conduction velocity was unaffected.

### 3.5. Case 5

A 62-year-old Caucasian man consulted our PCFC complaining of paresthesia and dysesthesia in the skin of the right hemithorax and the medial surface of the right lower limb. The patient developed severe COVID-19 pneumonia in November 2021, requiring admission to the intensive care unit. The personal history was otherwise uneventful, except for arterial hypertension. ENG showed reduced sensory nerve action potential amplitude consistent with sensory axonal neuropathy; nerve conduction velocity was normal. The histology examination revealed epidermal hyperkeratosis, acanthosis, and disappearance of rete ridges. Perivascular inflammation, characterized by plasma cells, eosinophils, and lymphomonocytic infiltrate, was also observed (Figure 5). S100-stain highlighted thinner and less numbers of nervous fibers in the superficial dermis and hypertrophic nerve bundles in the peri-adnexal area. The muco-polisaccarides deposition was also present.

### 3.6. Case 6

A 50-year-old female patient complained of acute hair loss initiating 1–3 months after the onset of SARS-CoV-2 infection. The patient reported intense trichodynia before and during the hair shedding. The trichoscopy showed typical telogen effluvium patterns. The trichogram showed a telogen rate greater than 20%. The pull test result was positive. 

A 4-mm punch biopsy of the scalp was performed. Histology showed hair follicles in anagen, telogen, and catagen phases. Median anagen/telogen ratio was 50%. No inflammatory infiltrate was present. There were various stages of dystrophic changes with evidence of keratin condensation as an initial event, followed by bulb atrophy up to fibrosis. The presence of hypertrophic nerve trunks revealed the hallmark of a hyper-neuritis condition (Figure 6).

The patient was prescribed daily oral supplementation with alpha-lipoic acid, biotin, calcium pantothenate, ferrous gluconate, and vitamin D3 for 3 months.

## 4. Discussion

Long COVID-19 syndrome is relatively common, affecting from 20 to 65% of patients after the acute phase of the disease. It is characterized by the persistence of symptoms such as fatigue, shortness of breath, and brain fog that cannot be explained by alternative diagnosis and that are not correlated with the severity of acute COVID-19. Peripheral neuropathy and neurosensory alterations are rarely reported in this condition. Neurologic involvement during SARS-CoV-2 infection may concern both central and peripheral nervous systems, being that the ACE2 receptor is widely expressed across neuronal cells [6,7,8,9,10]. These findings may be attributed to the prolonged effects of pro-inflammatory cytokines on brain biochemistry [11]. Alternatively, the virus could be able to directly infect and damage cells of the nervous system. 

Many pathogenetic mechanisms have been hypothesized for COVID-19 peripheral nervous system involvement. The immune-mediated effect of anti-GD1b autoantibodies and para-infectious damage of the peripheral nerves were described in Guillain-Barré and Miller-Fisher syndrome. Additional patho-mechanisms could be related to both the direct effect of SARS-CoV-2 or the indirect damage during the “cytokine storm”. It was also reported that SARS-CoV-2 may potentially access the nervous system and spread by anterograde and other routes of axonal transport, causing complex perceptive alterations [11,12,13].

Peripheral nervous system injury in the course of COVID-19 is likely underrecognized, and the incidence of its associated symptoms is likely to be underestimated, with reports ranging from 2.3% [14] to 5.8% [15]. Neuritic pain, paresthesia, and dysesthesia were reported in up to 8.9% of cases, whereas the mononeuritis multiplex was reported in 16% of COVID-19 patients [16]. In the latter condition, damage was attributed to vasculitis-associated neuropathy [17]. In a series of 103 patients hospitalized for SARS-CoV-2 infection, paresthesia and skin neurosensory symptoms have been diagnosed in six female COVID-19 patients corresponding to 5.8% of the study population [8] and a study by Krajewski et al. also described hyperestesia [9] with microstructural changes of peripheral sensory fibers [18,19,20,21].

Our study investigated cutaneous sNS in subjects with acute COVID-19, as well as with long COVID-19 syndrome, integrating clinical examination with histopathologic analysis and immunohistochemistry. Out of 142 patients examined, nine subjects (8.28%) complained of subjective symptoms compatible with neurosensory cutaneous involvement; six patients consented to skin biopsy. A shared feature in these patients was the subjective perception of burning pain, which was generalized over the cutaneous surface, did not follow specific dermatomeric distribution, and was apparently unrelated to objective skin lesions. The skin pain occurred in the late phases of COVID-19, and women were more frequently affected in line with the literature [8].

Paresthesia, alterations of pain and temperature sensation, are established sensory neurologic symptoms related to SARS-CoV-2 infection that may be due to an indirect mechanism related to virus-induced immune dysregulation [22,23]. Similarly, post-herpetic neuralgia is commonly related to nerve injury and alterations of sensory function [24].

Histopathology and immunohistochemistry in our study showed dermal nerve fiber hypertrophy manifesting as readily visible nerve bundles in the reticular dermis. This finding is typically described in prurigo nodularis due to increased Calcitonin gene-related peptide (CGRP) [25] and in the condition termed “dermal hyperneury” [26,27]. 

The neuropathic chronic pain in post-acute COVID-19 patients has been recently described by Ieremia et al., who reported loss of dermal neurosensory fibers in nine patients [28]. Compared to our findings of hypertrophy, fiber loss and inflammation could be related to a later phase of the disease, in which chronic inflammation leads to atrophy. On the ground of histopathology, we hypothesized that the sensory disturbance could be due to the malfunction of myelinated A-delta and unmyelinated C primary afferent neuronal fibers. Whether this is relatable to a direct action of the virus on terminal nerve fibers or to other indirect mechanisms still must be elucidated.

## 5. Conclusions

Paroxysmal attacks of burning and itching sensation related to COVID-19, and Long-COVID-19 as well, can be related to the damage of the neurosensory skin system, manifesting with uneven distribution, affecting the skin in a skip fashion irrespective of dermatomes. The symptoms have variable onsets in the course of SARS-CoV-2 infection and may persist for months, possibly accompanied by systemic manifestations such as asthenia, shortness of breath, and brain fog.

Histopathologic findings showed different alterations of small dermal fibers, depending on disease duration. Hypertrophy and inflammation may occur in the early phase of the disease, followed by atrophy and fiber loss in the chronic stage. 

Our experience in the management of COVID-19-related sensory neuropathic pain suggests that this condition is scarcely responsive to oral antihistamines. We have obtained satisfactory responses in our patients with low-dose oral gabapentin administered for three months. At a follow up visit after 6 months, all subjects reported remission of symptoms.

## Figures and Tables

**Figure 1 jcm-11-04474-f001:**
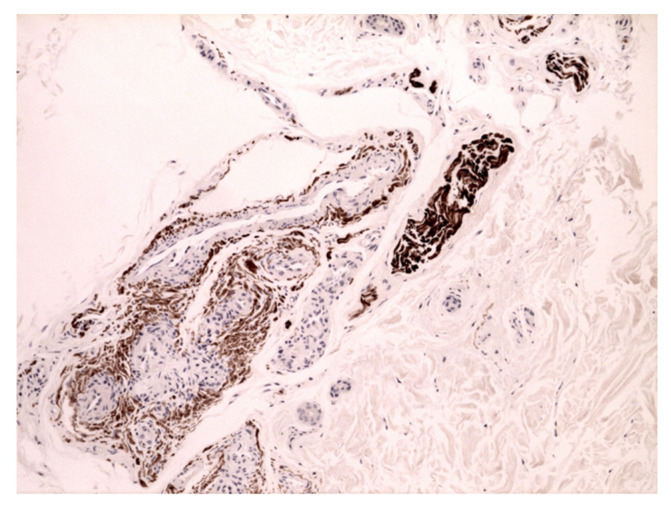
Neuropathic skin pain. Case 1: hypertrophic nerve bundle with numerous abnormal Schwann cells surrounding a glomus body. S100 immunostaining (40×).

**Figure 2 jcm-11-04474-f002:**
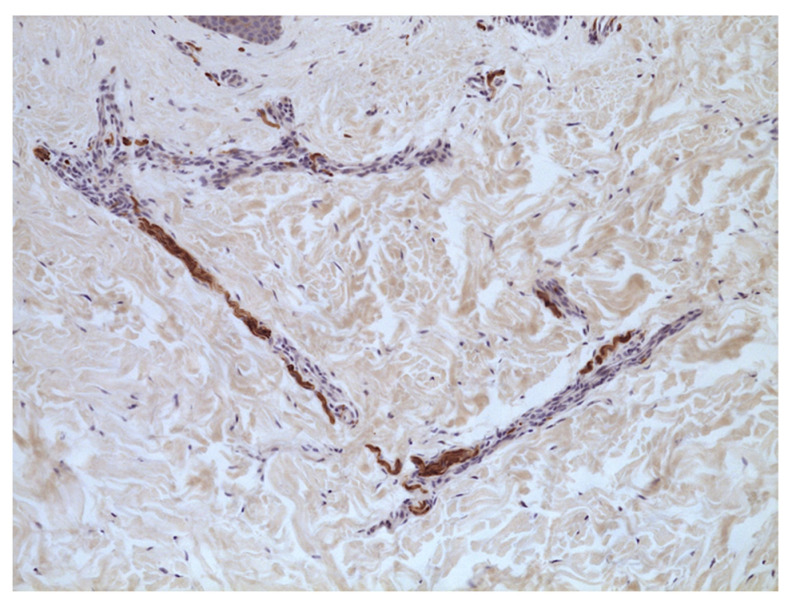
Neuropathic skin pain. Case 2: evidence of wavy hypertrophic nerve bundles. S100 immunostaining (40×).

**Figure 3 jcm-11-04474-f003:**
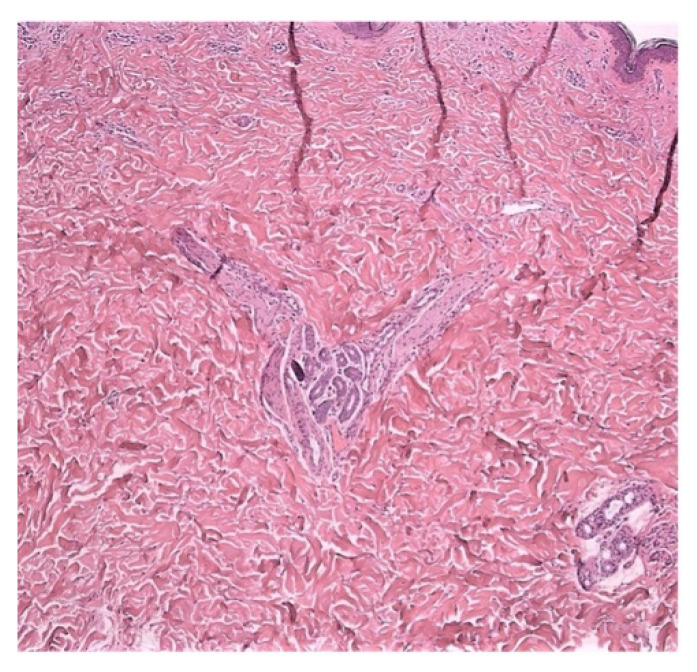
Neuropathic skin pain. Case 3: hypertrophic nerve bundles with wavy appearance. Hematoxylin and Eosin (40×).

**Figure 4 jcm-11-04474-f004:**
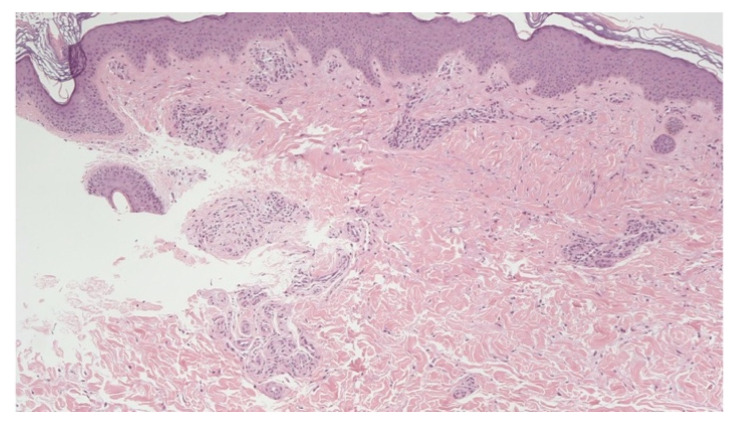
Neuropathic skin pain. Case 4: glomeruloid cluster of hypertrophic nerve bundles with basophilic mucin. No signs of inflammatory reaction were appreciated. Hematoxylin and Eosin (40×).

**Figure 5 jcm-11-04474-f005:**
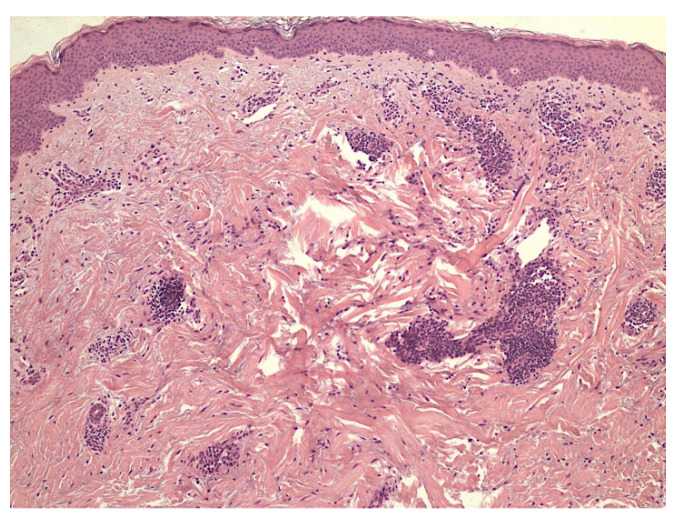
Neuropathic skin pain. Case 5: perivascular inflammation and slightly increased mucin content in the superficial dermis. Hematoxylin and Eosin (40×).

**Figure 6 jcm-11-04474-f006:**
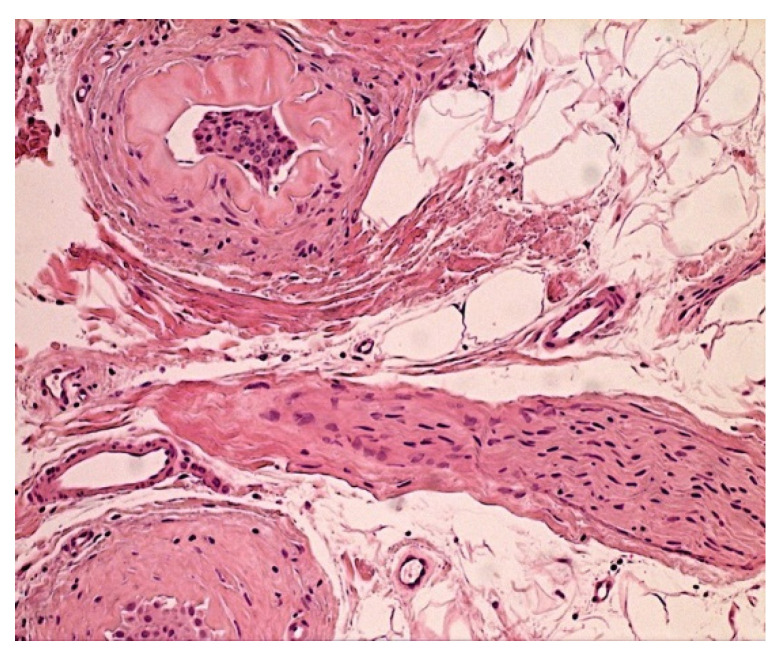
Neuropathic skin pain. Case 6: sclero-hyalinosis of the follicles, showing hypertrophic nerve trunk in the center. Hematoxylin and Eosin (200×).

**Table 1 jcm-11-04474-t001:** Demographic and histologic characteristics of six subjects with cutaneous subjective neurologic symptoms related to COVID-19.

Case	Sex	Age (Years)	Hypertrophic S100+ Peri-Neural Sheath Cells	Hypertrophy of the Nerve Branches	Increased Basophil Polysaccharide Matrix
1	F	49		x	
2	F	67	x	x	
3	M	64		x	
4	F	35		x	x
5	M	62	x	x	x
6	F	50		x	

Abbreviations: F, female; M, male; x, presence of the histologic feature.

## Data Availability

The data presented in this study are available on request from the corresponding author. The data are not publicly available due to privacy restrictions.

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
