# Peer review of "The Pathological Culprit of Neuropathic Skin Pain in Long COVID-19 Patients: A Case Series"

_jcm, 2022, doi:10.3390/jcm11154474_

Round 1

Reviewer 1 Report

The authors reported 6 cases of neuropathic skin pain in Long COVID-19 patients. The findings are interesting. I have some suggestions listed below.

1. The burning and stinging sensation in some of the patients reported is similar to that seen in patients with erythromelalgia. Please discuss the differences and similarities between erythromelalgia and the current reported condition.

2. Not all patients have checked for autoimmune profiles. Please provide these data since autoimmune-related conditions should be ruled out first before attributing to the Long COVID-19.

3. In addition to EMG, the nerve conduction velocity (NCV) test is also important to examine the neuropathic conditions. The authors should provide the results of the NCV tests.

4. How about the prognosis and the treatments for these patients? 

Author Response

Dear Reviewer,

we are grateful for your insightful and useful observations. Please find our point-by-point responses below:

  1. Thank you for your constructive comment. The main difference is that our patients referred disturbances of subjective perceptions associated to apparently normal skin.Though patient 1 showed bilateral palmar erythema and patient 2 complained palmar and plantar pain, the affected skin areas were not limited to the palmar-plantar regions but involved patches on all the cutaneous surface. Except for patient 1, that firstly manifested pseudo-chilblain palmar lesions related to COVID-19, all other patients complained burning pain without any skin involvementsuch as erythema or edema. Surprisingly the neuropathic disturbance manifested in apparently normal skin.Moreover, the “attacks” were not subjected to external triggers, like temperature variations or emotional stress, sweating or alcohol consumption, as in erythromelalgia. The following statement was added to case 1: “Though the patient presented with bilateral erythema of the palms, clinical suspect of erythromelalgia was excluded due to the absence of typical triggers for episodic pain and to the widespread burning sensation involving apparently normal distant skin.”
  2. Thank you for your suggestion: autoimmune profile was routinely checked in all subjects, with normal results. We reported results only for subject 1 in consideration of the skin findings of livedo reticularis and pseudo-chilblain that were highly suspicious for cutaneous vasculitis. To clarify this aspect, we have added the following sentence to the Methods: “Additionally, laboratory exams were routinely performed in all patients to rule out autoimmune conditions, with negative results.”
  3. Thank you. Electroneurography was indeed performed in two patients and nerve conduction velocity was assess and show to be normal (e.g., patient 5: motor NCV right peroneal 42 m/s and right ulnar 49 m/s; sensory NCV right ulnar 52 m/s). We added the following sentences to the description of case 4 and case 5, respectively: “ENG reported reduced sensory nerve action potential amplitude; nerve conduction velocity was unaffected.” and “ENG showed reduced sensory nerve action potential amplitude consistent with sensory axonal neuropathy; nerve conduction velocity was normal.”
  4. Thank you for your constructive comment. We added a statement in the Methods section to acknowledge that “all patients underwent follow up clinical visits after 3 and 6 six months.” The management was similar for all patients, excluding patient 6, for which we added a specific statement in the case description: “The patient was prescribed daily oral supplementation with alpha-lipoic acid, biotin, calcium pantothenate, ferrous gluconate, vitamin D3 for 3 months.” Management and outcome of other subjects is summarized in the conclusions: “Our experience in the management of COVID-19-related sensory neuropathic pain suggests that this condition is scarcely responsive to oral antihistamines. We have obtained satisfactory responses in our patients with low-dose oral gabapentin administered for three months. At a follow up visit after 6 months all subjects reported remission of symptoms.”

Reviewer 2 Report

Authors provide pathologic evidence of neurosensory skin system involvement in COVID-19 patients complaining subjective neurological symptoms affecting the skin. Out of 142 patients, six long COVID-19 cases complaining cutaneous subjective neurological symptoms assessed on NTSS-6 questionnaire, underwent histopathological and immunohistochemical analyses of skin areas affected by paroxysmal diffuse burning and itching sensation. Two patients also performed electroneurography examination. Histology investigation showed hypertrophic glomus vascular bodies with hypertrophic S100+ perineural sheath cells and adjacent hypertrophy of the nerve branches associated with increased basophil polysaccharide matrix. Electroneurography revealed disturbances of A-delta and C dermal neuronal fibers. Histopathology findings are consistent with hypertrophy of nerve endings, suggesting a condition like “dermal hyperneury”, a recently reported small nerve hypertrophy condition affecting sensory C fibers. Such neuropathic basis could explain dysesthesia experienced by the patients, as previously described in postherpetic neuralgia.

This paper provides a detailed clinical review of skin manifestations after COVID-19 infection. It gives some answers to questions that many dermatologists have. The small number of cases and the small number of histological evaluations may be the limitation of the present study, but the results are interesting as preliminary data. Further research is expected to be conducted in the future.

Author Response

Dear Reviewer,

thank you for your efforts on our paper and for your encouraging comments. Restrictive measures related to COVID-19 greatly impacted the access to skin biopsy procedures and consequently limited the size of our study sample. We plan on continuing our research on a broader number of subjects.

Round 2

Reviewer 1 Report

The manuscript has improved after revision. All my previous concerns have been well-addressed.